# The mitotic spindle protein CKAP2 potently increases formation and stability of microtubules

**Thomas S McAlear, Susanne Bechstedt***

Department of Anatomy and Cell Biology, McGill University, Montréal, Canada

**Abstract** Cells increase microtubule dynamics to make large rearrangements to their microtubule cytoskeleton during cell division. Changes in microtubule dynamics are essential for the formation and function of the mitotic spindle, and misregulation can lead to aneuploidy and cancer. Using in vitro reconstitution assays we show that the mitotic spindle protein Cytoskeleton-Associated Protein 2 (CKAP2) has a strong effect on nucleation of microtubules by lowering the critical tubulin concentration 100-fold. CKAP2 increases the apparent rate constant $k_a$ of microtubule growth by 50-fold and increases microtubule growth rates. In addition, CKAP2 strongly suppresses catastrophes. Our results identify CKAP2 as the most potent microtubule growth factor to date. These finding help explain CKAP2's role as an important spindle protein, proliferation marker, and oncogene.

## Editor's evaluation

This study reports the first substantial in vitro characterization of how the purified microtubule-associated protein CKAP2 interacts with microtubules. CKAP2 strongly promotes their nucleation and polymerization and suppresses their depolymerization. This work may stimulate more investigations into the physiological significance of these remarkably strong in vitro activities.

*For correspondence:
susanne.bechstedt@mcgill.ca

**Competing interest:** The authors declare that no competing interests exist.

## Introduction

During mitosis, cells build mitotic spindles to faithfully segregate their chromosomes. Assembling mitotic spindles requires a complete rearrangement of the microtubule cytoskeleton, which is driven by the concerted action of microtubule-associated proteins (MAPs) and motor proteins (*Kapoor, 2017*). Microtubule turnover increases significantly in mitosis resulting from increased microtubule nucleation (*Piehl et al., 2004*) in combination with shorter microtubule lifetimes (*Saxton et al., 1984*).

How exactly microtubule nucleation and growth in mitotic spindles is controlled and if all the major assembly factors have been identified remains an open question. XMAP215/chTOG and TPX2 have been implicated as two major factors in nucleation (*Roostalu et al., 2015*). In addition, XMAP215/chTOG family members are the only protein shown to speed up microtubule growth by more than a factor of 3 in in vitro assays (*Brouhard et al., 2008*).

Microtubules nucleate primarily from centrosomes as well as several centrosome-independent microtubule nucleation pathways aided by γ-tubulin ring complexes (γ-TURCs) (*Petry and Vale, 2015*). Notably, the γ-TURC only weakly increases microtubule nucleation and is thought to require activation of the complex itself or the help of MAPs like the microtubule polymerase XMAP215/chTOG or the nucleation factor TPX2 (*Consolati et al., 2020*; *Kollman et al., 2010*; *Moritz et al., 1995*; *Thawani et al., 2018*).

**Figure 1.** Cytoskeleton-Associated Protein 2 (CKAP2) is an intrinsically disordered protein that increases microtubule formation. (**a**) Schematic of mmCKAP2 protein domains and disorder prediction (*Dosztányi et al., 2005*; *Obradovic et al., 2003*). (**b**) Circular dichroism of 4.5 μM CKAP2. (**c**) Coomassie Blue-stained sodium dodecyl sulphate–polyacrylamide gel electrophoresis (SDS–PAGE) gel of 1 μg of purified recombinant CKAP2 constructs and 5 μg of purified tubulin. (**d**) Light scattering assay (turbidity) schematic and data following microtubule formation as apparent absorbance (Abs) over time for 8 μM tubulin with increasing concentrations of CKAP2-mNeonGreen (CKAP2-mNG) (*n* = 1). (**e**) Turbidity data for tubulin alone (blue) and addition of CKAP2-mNG (*n* = 3, mean ± standard deviation [SD]).

The online version of this article includes the following source data and figure supplement(s) for figure 1:

**Source data 1.** Protein purification gel.

**Figure supplement 1.** Cytoskeleton-Associated Protein 2 (CKAP2) is an intrinsically disordered protein that increases microtubule assembly.

Given the robustness of spindle assembly as well as the presence of microtubules in cells lacking chTOG (*Gergely et al., 2003*), TPX2 (*Aguirre-Portolés et al., 2012*), and γ-tubulin (*Strome et al., 2001*), it seems likely that additional factors can promote microtubule formation in spindles.

The Cytoskeleton-Associated Protein 2 (CKAP2) is an intrinsically disordered protein (*Figure 1a*) that localizes to centrosomes and mitotic spindles during cell division (*Seki and Fang, 2007*). CKAP2 expression and phosphorylation states are tightly regulated throughout the cell cycle. During G1 interphase, CKAP2 expression is at the detection level and increases at the onset of mitosis (*Seki and Fang, 2007*). CKAP2 is regulated by distinct phosphorylation events between different mitotic stages

(*Hong et al., 2009*). The anaphase-promoting complex (APC/C) marks CKAP2 through a conserved, N-terminal KEN-box motif (*Figure 1—figure supplement 1A*) for degradation at the end of mitosis to eliminate all CKAP2 from the newly formed daughter cells (*Hong et al., 2007*; *Seki and Fang, 2007*).

Knock-down of CKAP2 in cells interferes with proper spindle assembly and often results in multi-polar spindles and misaligned chromosomes (*Case et al., 2013*). Overexpression of CKAP2 promotes cancer formation (*Guo et al., 2017*; *Yu et al., 2015*) and correlates with severity of the disease (*Hayashi et al., 2014*). Consequently, CKAP2 is a marker for the diagnosis and prognosis of several types of cancer (*Hayashi et al., 2014*). CKAP2 has been previously described as a potential micro-tubule stabilizer in cells (*Jin et al., 2004*; *Tsuchihara et al., 2005*), though it is not known whether CKAP2 directly binds to microtubules or how it affects microtubule dynamics. We were intrigued by the cell biology of CKAP2, the severe spindle phenotypes, the clinical relevance, and the limited molecular understanding of this protein. Therefore, we wanted to investigate how CKAP2 impacts microtubule dynamics.

## Results

The primary amino acid sequence of CKAP2 is poorly conserved compared to other MAPs like TPX2 and chTOG and does not contain any known microtubule-binding domains (*Figure 1a*, *Figure 1—figure supplement 1A*). About half the protein consists of the CKAP2_C domain (IPR029197), that defines the CKAP2 family. Like TPX2, CKAP2 is predicted to be highly intrinsically disordered in solution, and both proteins are significantly enriched in lysines (CKAP2 ~10%, TPX2 ~13%), likely to facilitate interactions with the negatively charged C-terminal tails of tubulin.

About 60% of the CKAP2 protein sequence is predicted to be intrinsically disordered (*Figure 1A*). We performed circular dichroism (CD) to confirm this prediction. Indeed, we find 45% of CKAP2 to be disordered in solution (*Figure 1B*). Another 17% of the protein contains turns, 30% β-sheets, and 8% α-helices. Interestingly, secondary structure prediction methods as well as AlphaFold predict α-helices for the ordered CKAP2 segments (*Varadi et al., 2022*). In contrast, our CD data predominantly show β-sheets as the main secondary structure element for CKAP2 in solution. Notably, the measured disorder and secondary structure composition in solution for CKAP2 could change upon phosphorylation or interaction with tubulin, microtubules, or other binding partners as it has been shown for many intrinsically disordered proteins (*Uversky, 2019*).

To test for the ability of CKAP2 to influence microtubule polymer formation, we used recombinantly expressed full-length protein (*Figure 1C*) in a bulk light scattering assay (*Figure 1C*). At physiological tubulin levels (8 μM), increasing amounts of CKAP2 cause dose-dependent increase in light scattering and apparent absorbance (turbidity). At higher concentrations of CKAP2 turbidity continued to increase (*Figure 1E*) indicating that more microtubules are assembled faster. In addition, increased microtubule bundling likely contributes to the measured effect. Furthermore, CKAP2 increased turbidity at 4°C (*Figure 1—figure supplement 1D*), a temperature where tubulin does not form polymers and preformed microtubules depolymerize quickly in the absence of any stabilizers. Neither the mNeonGreen (mNG) nor the 6-His affinity tag had any impact on the ability of CKAP2 to aid microtubule formation (*Figure 1—figure supplement 1B,C*).

The bulk turbidity assay cannot distinguish between effects of CKAP2 on microtubule nucleation, growth, or stabilization. To test how CKAP2 enhances tubulin polymer formation, we reconstituted microtubule assembly in vitro using total internal reflection fluorescence (TIRF) microscopy (*Gell et al., 2010*). In this assay, we observe individual microtubules growing from guanosine-5′-[(α, β)-methyleno]triphosphate (GMPCPP)-stabilized microtubule 'seeds' (*Figure 2A*). At 8 μM tubulin, the lower end of estimated cellular tubulin concentrations and in the presence of CKAP2, observation of individual growth events was only possible at very low CKAP2 concentrations (≤50 nM) due to high levels of spontaneous nucleation beyond that point. The microtubule growth rate increased linearly with CKAP2 at these concentrations (*Figure 2B* and *Figure 2—figure supplement 1A*).

To measure microtubule growth across a wide range of CKAP2 concentrations and determine apparent tubulin on-and-off rates ($k_a$ and $k_d$) we reduced the tubulin concentration about 100-fold (to 50–300 nM) to mitigate spontaneous nucleation and observe individual microtubule growth events. The resulting growth curves (*Figure 2C* and *Figure 2—figure supplement 1B*) illustrate microtubule elongation at substantially lower tubulin concentrations than previously observed (*Wieczorek et al., 2015*). Microtubules polymerized with an apparent assembly rate constant ($k_a$) of 2.6 ± 0.06 dimers

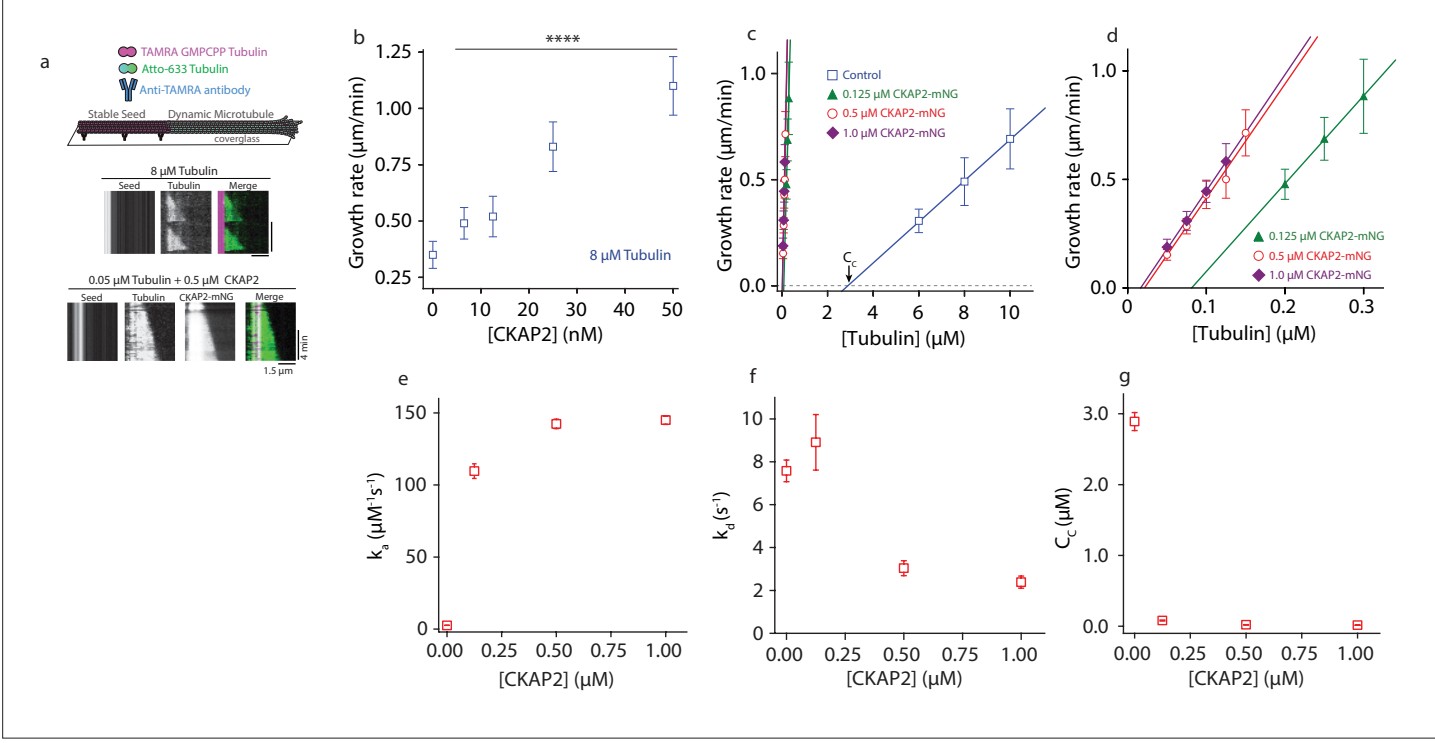

**Figure 2.** Cytoskeleton-Associated Protein 2 (CKAP2) lowers the critical concentration of microtubule growth and speeds up microtubule assembly rates. (**a**) Dynamic microtubule growth assay schematic (upper panel). Microtubule dynamics for tubulin control and in the presence of CKAP2-mNeonGreen (mNG) are analyzed from space–time plots (kymographs, lower panel). (**b**) Plot of microtubule growth rates as a function of CKAP2-mNG concentration with 8 µM tubulin. Plotted as mean ± standard deviation (SD; $n$ = 67, 24, 70, 67, 46 from 1 to 2 replicates) Tukey's test and one-way analysis of variance (ANOVA) used to compare mean values of raw data. ****$p \leq 0.0001$. (**c**) Plot of microtubule growth rates as a function of tubulin concentration for control (no CKAP2) and in the presence of CKAP2-mNG. Plotted as mean ± SD (blue; $n$ = 272, 601, 347 from three replicates), 0.125 µM CKAP2-mNG (green; $n$ = 146, 85, 91 from two replicates), 0.5 µM CKAP2-mNG (red; $n$ = 48, 92, 91, 85, 55 from two replicates), and 1 µM CKAP2-mNG (purple; $n$ = 68, 134, 143, 115 from two replicates). (**d**) Enlargement at low tubulin concentrations of microtubule growth rate plot from (**c**). (**e**) Plot of the apparent on-rate constant ($k_a$) as a function of CKAP2-mNG concentration determined from linear regression fit of growth rates versus tubulin concentration. Error bars represent standard error (SE) of fit. (**f**) Plot of the apparent off-rate constant ($k_d$) as a function of CKAP2-mNG concentration determined from linear regression fit of growth rates versus tubulin concentration. Error bars represent SE of fit. (**g**) Plot of the apparent critical concentration ($C_c$) as a function of CKAP2-mNG concentration determined from linear regression fit of growth rates versus tubulin concentration. Error bars represent SE of fit.

The online version of this article includes the following source data and figure supplement(s) for figure 2:

**Source data 1.** The data and analysis for microtubule growth rates for different concentrations of Cytoskeleton-Associated Protein 2 (CKAP2).

**Figure supplement 1.** Cytoskeleton-Associated Protein 2 (CKAP2) lowers the critical concentration of microtubule growth and speeds up microtubule assembly rates.

µM⁻¹ s⁻¹ in controls and 142 ± 3.3 dimers µM⁻¹ s⁻¹ in the presence of 500 nM CKAP2, representing a 54-fold increase in $k_a$ (**Figure 2D**). By comparison, members of the XMAP215/chTOG family of microtubule polymerases achieve about fivefold maximal increase in $k_a$ (**Brouhard et al., 2008**). We observed a 42-fold increase in $k_a$ for 125 nM CKAP2. Increasing the CKAP2 concentration beyond 500 nM did not further increase growth rates. We observe a smaller effect (~threefold) on the tubulin off-rate $k_d$ (**Figure 2E**).

From the growth curves in **Figure 2C** and the spontaneous nucleation we observe at quasi-physiological concentrations of CKAP2 and tubulin, it is apparent that CKAP2 is able to shift the critical concentration ($C_c$) for microtubule elongation into the low nanomolar range (from 2.89 ± 0.12 to 0.02 ± 0.002 µM with 500 nM CKAP2) (**Figure 2F**).

To characterize the nucleation behaviour of microtubules we measured the probability of a seed to nucleate a microtubule over time (**Wieczorek et al., 2015**; **Figure 3A**). At quasi-physiological tubulin levels microtubules nucleated from seed templates faster in the presence of as ≥25 nM CKAP2 (**Figure 3B**). When we explored a wider range of CKAP2 and tubulin concentrations, we found a very

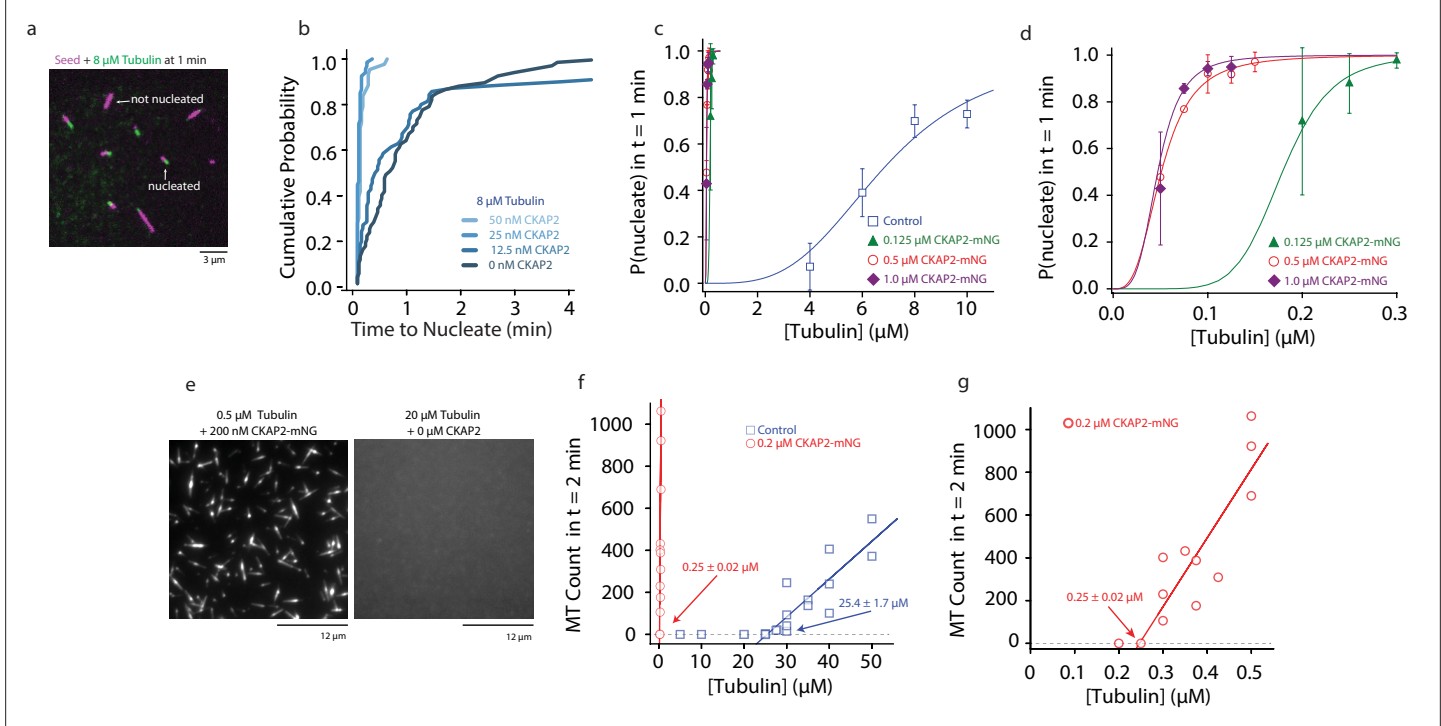

**Figure 3.** Cytoskeleton-Associated Protein 2 (CKAP2) increases templated and spontaneous microtubule nucleation. (**a**) Representative field of view of guanosine-5'-[(α,β)-methyleno]triphosphate (GMPCPP)-stabilized microtubule seeds nucleating microtubules from 8 µM tubulin. (**b**) Plot of the cumulative probability distributions for time to nucleate as a function of CKAP2-mNeonGreen (mNG) concentration (n = 67, 70, 67, 46 from two replicates). (**c**) Plot of probability of GMPCPP seed to be nucleated within 1 min for control (blue; n = 102, 271, 115, 184 from ≥2 replicates), 0.125 µM CKAP2-mNG (green; n = 164, 213, 166 from two replicates), 0.50 µM CKAP2-mNG (red; n = 179, 216, 145, 126, 145 from two replicates), and 1 µM CKAP2-mNG (purple; n = 112, 186, 182, 148 from two replicates). Plotted as mean ± standard deviation (SD). Data were fit to a hill function forced to start at y = 0 and end at y = 1 ($y = START + (END - START) \times x^n/(k^n + x^n)$). Tubulin concentrations for half-maximal microtubule growth, $C$, and steepness of fit, $s$, for control ($C = 6.85 \pm 0.48$ µM, $s = 3.37 \pm 0.87$), 0.125 µM CKAP2-mNG ($C = 0.18 \pm 0.01$ µM, $s = 6.96 \pm 1.39$), 0.5 µM CKAP2-mNG ($C = 0.05 \pm 0.01$ µM, $s = 3.10 \pm 0.20$), and 1 µM CKAP2-mNG ($C = 0.05 \pm 0.01$ µM, $s = 3.79 \pm 0.85$). (**d**) Enlargement at low tubulin concentrations of probability to nucleate plot from (**c**). (**e**) Representative field of view of microtubules spontaneously nucleated in the absence of templates. (**f**) Quantification of spontaneous nucleation and linear fit showing a critical tubulin concentration for microtubule nucleation of 25.4 ± 1.7 µM for tubulin alone (blue; n = 20) and 0.25 ± 0.02 µM in the presence of 0.2 µM CKAP2 and 0.25 ± 0.02 µM (red; n = 14). (**g**) Enlargement at low tubulin concentrations of spontaneous nucleation in the presence of 0.2 µM CKAP2 from (**e**).

The online version of this article includes the following source data and figure supplement(s) for figure 3:

**Figure supplement 1.** Cytoskeleton-Associated Protein 2 (CKAP2) increases templated and spontaneous microtubule nucleation.

**Source data 1.** The data and analysis for templated and spontaneous microtubule nucleation for different concentrations of Cytoskeleton-Associated Protein 2 (CKAP2).

strong promotion of templated nucleation by CKAP2 even at 100-fold lower tubulin concentrations compared to controls (**Figure 3C** and **Figure 3—figure supplement 1A, E**). The tubulin concentration for the half-maximal probability to nucleate within 1 min is reduced from 6.85 ± 0.48 µM in controls to 0.05 ± 0.01 µM for 0.5 µM CKAP2-mNG. CKAP2 facilitates templated nucleation from seeds at tubulin concentrations as low as 50 nM. Further, we quantified the spontaneous nucleation, that is in the absence of templates, in our microscopy assay by counting the number of microtubules formed after 2 min (**Figure 3D**; **Roostalu et al., 2015**). We found that CKAP2 dramatically increases spontaneous nucleation with a shift in critical tubulin concentration from 25.4 ± 1.7 to 0.25 ± 0.02 µM (**Figure 3E** and **Figure 3—figure supplement 1B**). CKAP2 therefore promotes both templated and spontaneous nucleation. In the presence of CKAP2, the $C_c$ of spontaneous nucleation, as well as nucleation from templates, are lowered up to 100-fold.

A hallmark of microtubule dynamic instability is their capacity to spontaneously switch from growth to shrinkage, a behaviour termed catastrophe (**Mitchison and Kirschner, 1984**). When microtubules were grown in the presence of CKAP2, catastrophe levels were severely reduced (**Figure 4A**). We

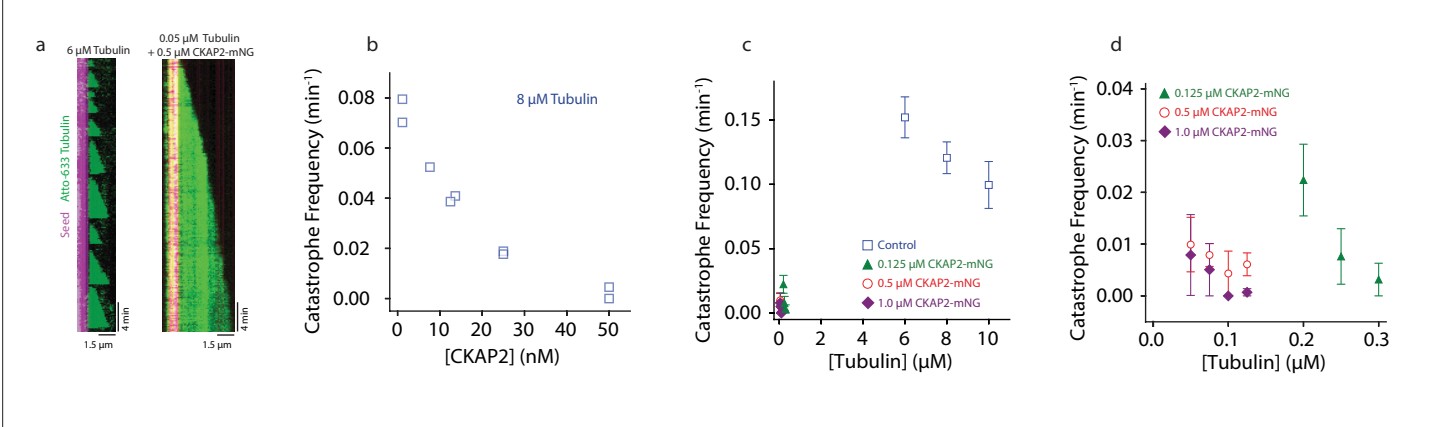

**Figure 4.** Cytoskeleton-Associated Protein 2 (CKAP2) suppresses catastrophe. (**a**) Kymographs of representative microtubule growth in the absence and presence of CKAP2-mNeonGreen (mNG). (**b**) Plot showing the microtubule catastrophe frequencies as a function of CKAP2-mNG concentration. Each point represents the catastrophe frequency for an individual channel ($n$ (MT growth events) = 30, 37, 24, 31, 49, 30, 35, 19, 25, number of catastrophes = 26, 34, 16, 16, 26, 8, 5, 0, 1 for channels CKAP2 concentrations 0, 0, 6.5, 12.5, 12.5, 25, 25, 50, 50 nM). (**c**) Plot showing the microtubule catastrophe frequencies as a function of tubulin concentration for control (blue; $n$ (MT growth events) = 217, 676, 502, number of catastrophes = 217, 676, 502 from ≥2 replicates), 0.125 μM CKAP2-mNG (green; $n$ (MT growth events) = 133, 222, 166, number of catastrophes = 21, 10, 4 from two replicates), 0.50 μM CKAP2-mNG (red; $n$ = 161, 232, 95, 135, number of catastrophes = 9, 17, 2, 6 from two replicates), and 1 μM CKAP2-mNG purple ($n$ (MT growth events) = 85, 171, 182, 148, number of catastrophes = 5, 7, 0, 1 from two replicates). Plotted as mean ± standard deviation (SD). (**d**) Enlargement at low tubulin concentrations of catastrophe frequency plot (**b**).

The online version of this article includes the following source data and figure supplement(s) for figure 4:

**Source data 1.** The data and analysis for catastrophe frequency measurements.

**Figure supplement 1.** Cytoskeleton-Associated Protein 2 (CKAP2) suppresses catastrophe.

quantified this effect and determined that CKAP2 lowers catastrophe frequencies in a concentration-dependent manner, reducing them to near zero at 50 nM CKAP2 (*Figure 4B*). In the presence of higher concentrations of CKAP2, catastrophe frequency was at least an order of magnitude lower than in controls (*Figure 4C* and *Figure 4—figure supplement 1A, B, C*). The only other family of MAPs with a comparable effect on microtubule catastrophes are the cytoplasmic linker-associated proteins (CLASPs) (*Aher et al., 2018*; *Lawrence et al., 2018*; *Majumdar et al., 2018*).

A prediction for a potential cytoskeletal growth factor is the interaction with both the growing filament end and free monomers, as it has been shown for members of the chTOG/XMAP215 family for microtubules (*Ayaz et al., 2014*) and formins for actin (*Kovar et al., 2006*). Using size exclusion chromatography, we do not find any significant interaction of CKAP2 and tubulin in solution (*Figure 5A*). In contrast, CKAP2 is able to recruit soluble tubulin to microtubules. Microtubule lattice-bound CKAP2 recruits free Atto-663 tubulin to GMPCPP microtubule seeds (*Figure 5b, c*). CKAP2 therefore possesses a distinct microtubule lattice as well as likely one or multiple tubulin-binding sites. Recruitment of tubulin by lattice-bound CKAP2 could therefore play a role in promoting microtubule nucleation and growth.

Many of the proteins promoting microtubule nucleation and growth possess high-affinity-binding sites at the plus end of dynamic microtubules (e.g. XMAP215/chTOG, DCX, TPX2). Here, XMAP215/chTOG is thought to help the addition of tubulin dimers to the growing end (*Howard and Hyman, 2007*). To determine if CKAP2 interacts with the growing microtubule end, we performed our TIRF microscopy assay at subsaturating CKAP2 concentrations. Indeed, at low protein concentrations (6.5 nM), CKAP2 tracks microtubule ends (*Figure 5D*). At higher concentrations (12.5 nM) end tracking is obscured by lattice-binding reminiscent of end tracking behaviour observed for the microtubule nucleation factors DCX and TPX2 (*Figure 5D*; *Bechstedt and Brouhard, 2012*, *Roostalu et al., 2015*). CKAP2 shows higher affinity to the dynamic microtubule lattice over the GMPCPP lattice, unlike TPX2 (*Roostalu et al., 2015*). Similar to TPX2 (*Roostalu et al., 2015*) and DCX (*Bechstedt and Brouhard, 2012*), CKAP2 recognizes microtubule curvature (*Figure 5e, f*), which is thought to correlate with the high-affinity-binding site of these proteins at the microtubule end (*Bechstedt et al., 2014*; *Roostalu et al., 2015*). In contrast, members of the XMAP215/chTOG family display robust autonomous end

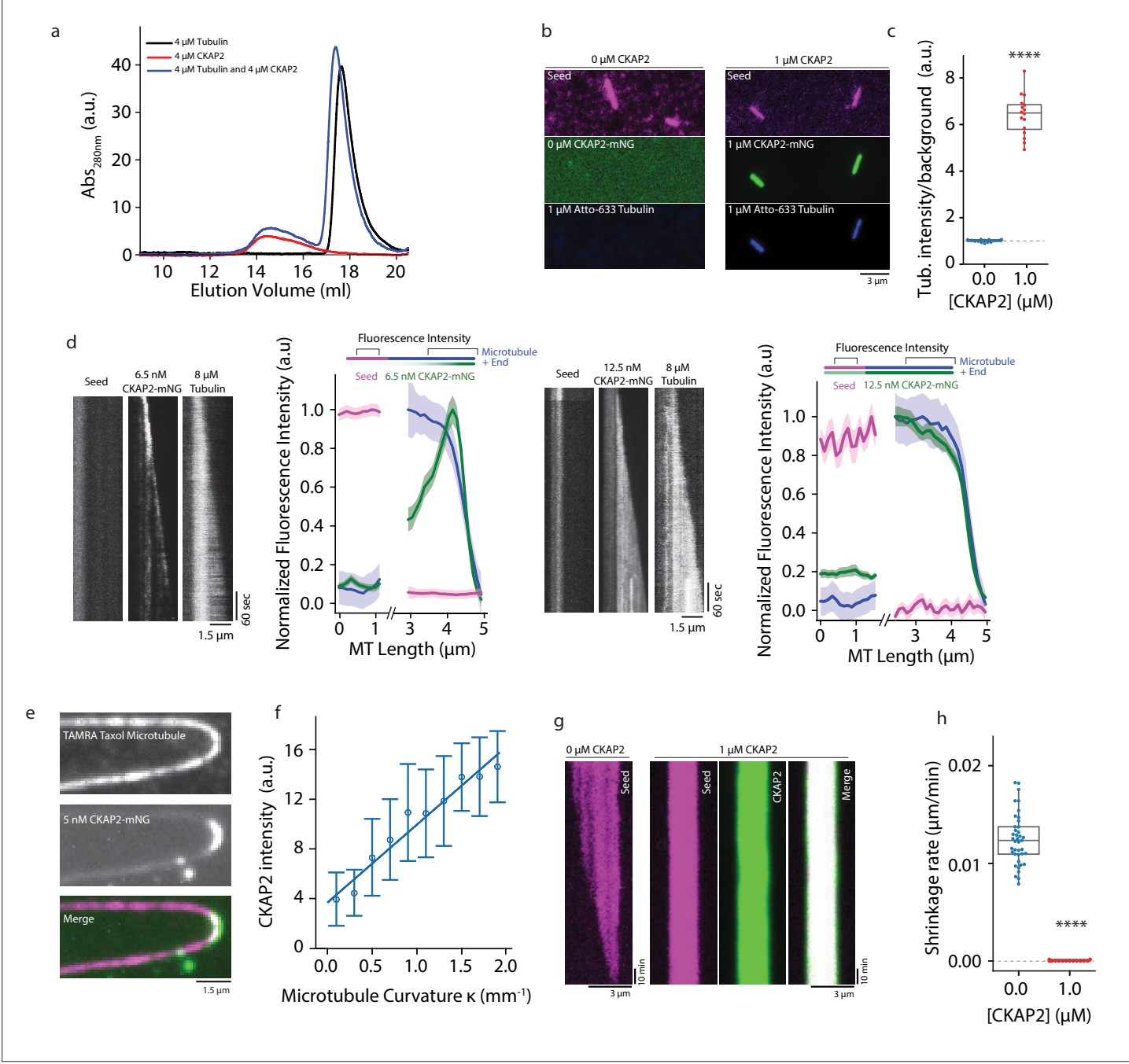

**Figure 5.** Cytoskeleton-Associated Protein 2 (CKAP2) recruits tubulin, recognizes lattice curvature, but does not catalyze microtubule depolymerization. (**a**) Size exclusion chromatography traces for 4 μM CKAP2, 4 μM tubulin, and a combination of both. (**b**) Field of view showing 1 μM CKAP2-mNeonGreen (mNG) recruiting Atto-633 tubulin to guanosine-5'-[(α,β)-methyleno]triphosphate (GMPCPP) seeds. (**c**) Box plot of tubulin intensity on the GMPCPP lattice over background for control and 1 μM CKAP2-mNG (blue; n = 17 from one representative channel) and 1 μM CKAP2-mNG (red; n = 16 from one representative channel). Tukey's test and one-way analysis of variance (ANOVA) used to compare mean values of raw data. ****p ≤ 0.0001. (**d**) Kymographs representative of 6.5 and 12.5 nM CKAP2-mNG on a growing microtubule seed and plus end. Plot of normalized average fluorescence intensity of CKAP2-mNG, microtubule seed, and dynamic microtubule lattice along microtubule seed and + end (6.5 nM) (n = 42 seed, 54 tip), 12.5 nM (n = 17 seed, 26 tip). Plotted as normalized background subtracted average ± standard error (SE). (**e**) Field of view of 5 nM CKAP2-mNG preferentially binding to curved regions of tetramethylrhodamine (TAMRA)-labelled paclitaxel-stabilized microtubules. (**f**) Plot of CKAP2-mNG intensity versus curvature (κ) measured using Kappa (***Mary and Brouhard, 2019***) (n = 73 from two replicates. Plotted as binned mean ± standard deviation [SD]). (**g**) GMPCPP seed depolymerization kymographs representative of n = 41 (control) and 16 (1 μM CKAP2-mNG). (**h**) Box plot displaying the shrinkage rate of GMPCPP microtubule seeds for control (blue; n = 41 from one representative channel) and in the presence of 1 μM CKAP2-mNG (red; n = 16

*Figure 5 continued on next page*

*Figure 5 continued*

from one representative channel). Tukey's test and one-way ANOVA used to compare mean values of raw data. ****p ≤ 0.0001.

The online version of this article includes the following source data and figure supplement(s) for figure 5:

**Source data 1.** The data and analysis for tubulin recruitment by Cytoskeleton-Associated Protein 2 (CKAP2) and shrinkage rate for guanosine-5′-[(α,β)-methyleno]triphosphate (GMPCPP) microtubules in the presence and absence of CKAP2.

**Figure supplement 1.** Cytoskeleton-Associated Protein 2 (CKAP2) recruits tubulin, recognizes lattice curvature, but does not catalyze microtubule depolymerization.

tracking at all physiologically relevant concentrations and do not recognize lattice curvature (*Brouhard et al., 2008*; *Roostalu et al., 2015*).

We next tested if CKAP2 can promote the removal of tubulin dimers from the lattice in the absence of tubulin. Catalyzing this reverse reaction of microtubule growth is a hallmark of *bona fide* microtubule polymerases of the XMAP215/chTOG family (*Brouhard et al., 2008*). We did not observe any depolymerization over ~10 hr (*Figure 5g, h* and *Figure 5—figure supplement 1A*) when we incubated GMPCPP-stabilized microtubules with CKAP2 at concentrations as high as 1 µM, while controls without CKAP2 displayed an expected depolymerization rate of about 0.013 ± 0.003 µm/min (*Figure 5g, h*). CKAP2 does not catalyze the reverse reaction. On the contrary, CKAP2 strongly reduces the tubulin off-rate.

## Discussion

Our results show that CKAP2 has a strong effect on microtubule growth, nucleation, and stabilization in vitro.

Cellular concentrations of CKAP2 in *Xenopus* eggs have been measured at about 60 nM in conjunction with 8.5 µM tubulin (*Wühr et al., 2014*). At these quasi-physiological CKAP2 and tubulin concentrations (assuming 60 nM and 8 µM) CKAP2 has a strong effect on the microtubule growth rate (~fivefold) in our assays (*Figure 1B*). Further, templated nucleation is almost instantaneous (*Figure 3B*) and spontaneous nucleation is detectable. Catastrophe frequency is approaching zero at this concentration regime (*Figure 4B*). Therefore, in cells, CKAP2 could increase microtubule nucleation, growth rates, and suppress catastrophe. Interestingly, at quasi-cellular CKAP2 concentrations, the effects on microtubule dynamics in our in vitro assays are not fully saturated. Hence, cells may be susceptible to increased levels of CKAP2 as they have been observed in several types of cancer.

Notably, we purify CKAP2 from *E. coli* likely in an unphosphorylated state. In cells, sharp phosphorylation events are observed for CKAP2 between different mitotic stages which could modify and shift the effect of CKAP2 on different microtubule dynamics parameters (*Hong et al., 2008*).

At full biochemical capacity of CKAP2, we observed a 100-fold increase in spontaneous and templated nucleation as well as a 54-fold increase in apparent tubulin on rate $k_a$, and a complete suppression of microtubule catastrophe. Combined, these effects result in CKAP2 being the most potent microtubule assembly factor characterized to date.

Mechanistically, CKAP2 combines characteristics of microtubule nucleators (TPX2, DCX) as well as polymerases (XMAP215/chTOG) and anti-catastrophe factors (CLASPs). Like TPX2 and DCX, CKAP2 aids spontaneous nucleation. All three proteins display end tracking behaviour at low concentrations as well as lattice curvature recognition. These characteristics seem to be a hallmark of microtubule nucleation factors and might point towards the structural nature of the nascent nucleus. Similar to TPX2, CKAP2 is largely intrinsically disordered and dampens tubulin off-rates and microtubule dynamicity (*Reid et al., 2016*; *Roostalu et al., 2015*). A potential mechanism for nucleation by CKAP2 is the stabilization of the nascent microtubule nucleus (*Roostalu and Surrey, 2017*). Similar to the CLASP anti-catastrophe factors, CKAP2 lowers the catastrophe frequency to virtually zero (*Aher et al., 2018*; *Lawrence et al., 2018*; *Majumdar et al., 2018*), allowing for continued microtubule growth even at nanomolar concentrations of tubulin.

Other MAPs that stabilize the microtubule lattice like DCX (*Moores et al., 2006*), TPX2 (*Reid et al., 2016*), and CLASPs (*Lawrence et al., 2018*; *Yu et al., 2016*), as well as stabilizing drugs like paclitaxel (*Zanic et al., 2013*) or Guanosine-5′-triphosphate (GTP) analogues like GMPCPP (*Hyman et al., 1992*), do not severely impact the microtubule growth curve. Therefore, lattice stabilization alone cannot account for the severe impact of CKAP2 on the apparent tubulin on rate $k_a$.

Unlike TPX2, DCX, and CLASPs, CKAP2 also acts, in part, like a polymerase. Similar to known polymerases of the XMAP215/chTOG family, microtubule-bound CKAP2 recruits tubulin and increases the association rate constant ($k_a$) at the microtubule end (*Brouhard et al., 2008*; *Gard and Kirschner, 1987*). In contrast to XMAP215/chTOG family members, CKAP2 is able to push microtubule growth to the physical limit as defined by diffusion of the tubulin dimer to the growing microtubule end (*Odde, 1997*; *Zanic et al., 2013*). Unlike a polymerase, CKAP2 does not catalyze the reverse reaction of tubulin polymerization in the absence of tubulin substrates. CKAP2 either does not operate as a *bona fide* polymerase, or the depolymerase activity is masked by the lattice stabilization activity of CKAP2. In contrast to XMAP215/chTOG, CKAP2 does not interact with GDP tubulin in size exclusion chromatography experiments.

In many ways, microtubule growth in the presence of CKAP2 resembles that of microtubules grown in the presence of the slowly hydrolysable GTP analogue, GMPCPP. With GMPCPP, microtubules nucleate at nanomolar concentrations of tubulin and are relatively stable (*Desai and Mitchison, 1997*; *Hyman et al., 1992*). However, GMPCPP does not increase apparent on rate $k_a$, like CKAP2.

How does CKAP2 combine multiple functionalities? The protein is about one third of the size of XMAP215/chTOG, about half the size of CLASPs, and is lacking any known microtubule or tubulin-binding domains. We believe that the highly disordered nature of CKAP2 (*Figure 1a, b*) contributes to its ability to dynamically interact with the microtubule lattice as well as tubulin and impact microtubule assembly to an extreme extent. The advantage of disordered over folded domains is the absence of a structured, tight binding conformation to a substrate or intermediate state. In case of a microtubule growth factor, lack of a fixed structure would allow for fast molecular recognition of the tubulin dimer as well as for rapid unbinding, that is letting go of the dimer when incorporated (*Uversky and Dunker, 2013*). Binding of CKAP2 to microtubules likely causes the adoption of a secondary structure of part of the protein. Such structural rearrangements could explain the lack interaction with tubulin in solution in contrast to lattice-bound CKAP2. Structural as well as structure–function studies are needed to determine the interaction domains with microtubules and tubulin and provide us with further mechanistic insight.

Even small changes in microtubule assembly rates are correlated with increased levels of chromosomal instability (CIN) in cells (*Ertych et al., 2014*). CIN can provide the high adaptation capability necessary for tumor initiation and progression. Correspondingly, it has been shown that reducing microtubule growth rates in cancer cells by chemical or genetic manipulation suppresses CIN (*Ertych et al., 2014*). If the capacity of CKAP2 to promote microtubule nucleation and growth we find in vitro even partially translates into cells, we would expect significant changes in microtubule dynamics upon reduced or increased CKAP2 protein levels. Upregulation of a potent growth factor like CKAP2 as observed in many cancers could therefore be directly responsible for CIN, and subsequently for aneuploidy, and malignancies.

Cells need to tightly regulate the expression and activity of a potent microtubule growth factor like CKAP2. Known regulatory measures include rapid degradation by APC (*Seki and Fang, 2007*) as well as distinct phosphorylation and dephosphorylation events in between different stages of the cell cycle (*Hong et al., 2008*; *Hong et al., 2009*). Phosphorylation could alter the effects of CKAP2 on microtubule dynamics and tune CKAP2 functionality between nucleation, stabilization, and growth.

Together, our findings identify the mitotic spindle protein CKAP2 as the most potent microtubule growth factor to date. The protein displays unprecedented impact on microtubule growth, nucleation, and catastrophe. Our results suggest an explanation for the observed CKAP2 knock-down spindle phenotypes and the oncogenic potential through the misregulation of microtubule dynamics.

## Materials and methods
### Protein expression and purification
The coding sequence for full-length mouse CKAP2 protein (Uniprot accession # Q3V1H1) was PCR amplified from cDNA from wild-type mouse testes (a gift from Alana Watt's Lab) using PfuX7 (*Nørholm, 2010*) polymerase and inserted into a modified pHAT vector containing an N-terminal 6xHis-tag with and without a carboxy-terminal mNeonGreen (Allele Biotech) followed by a Strep-tag II (*Bechstedt and Brouhard, 2012*). Full-length CKAP2 without a 6xHis-tag still containing a carboxy-terminal

mNeonGreen followed by a Strep-tag II was codon optimized for *E. coli* and synthesized into a pET21 vector (Twist Bioscience). Pasmids are available from Addgene or the corresponding author.

For recombinant protein expression, BL21(DE3) *E. coli* containing protein expression vectors were grown to OD 0.6 at 37°C, and expression was induced using 0.5 mM Isopropyl ß-D-1-thiogalactopyranoside (IPTG) at 18°C for 16 hr. Bacterial pellets were harvested by centrifugation and resuspended in Buffer A (50 mM $Na_2HPO_4$, 300 mM NaCl, 4 mM imidazole, pH 7.8). Cells were lysed using a French press (EmulsiFlex-C5, Avestin). Constructs containing both His and Strep tags were purified using gravity flow columns containing His60 Ni-NTA resin (Clontech) followed by Streptactin affinity chromatography (IBA Lifesciences, Germany). Purified CKAP2 was eluted with BRB80 (80 mM PIPES–KOH, pH 6.85, 1 mM EGTA, 1 mM $MgCl_2$) or Tris–HCl (100 mM Tris–HCl pH 7.5, 150 mM NaCl, 1 mM EDTA) containing 2.5 mM desthiobiotin and 10% glycerol. For the CKAP2 no His construct, lysed protein was purified by cation exchange using a 1 ml HiTrap SP HP (GE Healthcare) in protein buffer (50 mM Tris–HCl, pH 7.0, 2 mM $MgCl_2$, 1 mM EGTA, and 10%) and eluted with a salt gradient from 50 to 400 mM NaCl. Protein was further purified using Streptactin affinity chromatography as per the other constructs. Purified CKAP2 was always used fresh for future experiments. Protein concentration was determined by absorbance at 288 or 506 nm with a DS-11 FX spectrophotometer (DeNovix, Inc).

Tubulin was purified from bovine brains as previously described (*Ashford and Hyman, 2006*) with the modification of using Fractogel EMD SO3- (M) resin (Millipore-Sigma) instead of phosphocellulose. Tubulin was labelled using Atto-633 NHS-Ester (ATTO-TEC) and tetramethylrhodamine (TAMRA, Invitrogen) as described (*Hyman et al., 1991*). An additional cycle of polymerization/depolymerization was performed before use. Protein concentrations were determined using a DS-11 FX spectrophotometer (DeNovix, Inc).

## CD spectroscopy

CD spectra were collected for purified unlabelled CKAP2 (4.5 µM) in Tris–HCl buffer (100 mM Tris–HCl, pH 7.5, 100 mM NaCl, 1 mM EDTA) using a Chirascan CD spectrometer (Applied Photophysics, Leatherhead, Surrey, UK). Individual CD spectra were collected at room temperature from 180 to 260 nm, using a 0.2 µm path-length cuvette and a collection time of 0.5 s/nm. The final experimental spectrum represents an average of 12 scans of the CKAP2 sample reference corrected by the average of 12 similarly collected buffer scans. To determine the secondary structural content of CKAP2, this spectrum was fitted against a library of 185–260 nm base spectra with the CONTINLL deconvolution algorithm using OLIS SpectralWorks software (On-line Instrument Systems, Bogart, GA).

## Microscopy

The microscope setup uses a Zeiss Axiovert Z1 microscope chassis and a 100 ×1.45 NA Plan-apochromat objective lens. TIRF was achieved by coupling 488/561/637 nm lasers to an iLas2 targeted laser illumination system (BioVision, Inc) equipped with 360° TIRF. The objective was heated to 35°C with a CU-501 Chamlide lens warmer (Live Cell Instrument). Data were also acquired with a customized Zeiss Axio Observer seven equipped with a Laser TIRF III and 405/488/561/638 nm lasers, Alpha Plan-Apo 100 ×/1.46Oil DIC M27, and Objective Heater 25.5/33 S1. Images on both systems were recorded on a Prime 95B CMOS camera (Photometrics) with a pixel size of 107 nm.

## Dynamic microtubule growth assay

To visualize dynamic microtubules, we reconstituted microtubule growth off of GMPCPP double stabilized microtubule 'seeds' (*Gell et al., 2010*). In short, cover glass was cleaned in acetone, sonicated in 50% methanol, sonicated in 0.5 M KOH, exposed to air plasma (Plasma Etch) for 3 min, then silanized by soaking in 0.2% dichlorodimethylsilane in *n*-heptane. 5 µl flow channels were constructed using two pieces of silanized cover glasses (22 × 22 and 18 × 18 mm) held together with double-sided tape and mounted into custom-machined cover slip holders. GMPCPP seeds were prepared by polymerizing a 1:4 molar ratio of TAMRA labelled:unlabelled tubulin in the presence of GMPCPP (Jena Biosciences) in two cycles, as described by *Gell et al., 2010*. Channels were first incubated with anti-TAMRA antibodies (Invitrogen) and then blocked with 5% Pluronic F-127. Flow channels were washed three times with BRB80 before incubating with GMPCPP seeds. On each day of experiments tubes of unlabelled and Atto-633 labelled tubulin was thawed and mixed at a 1:17 molar ratio and then subaliquoted and

refrozen in liquid nitrogen. For consistency in microtubule growth dynamics, one subaliquot of tubulin was used for each experiment. Microtubule growth from GMPCPP seeds was achieved by incubating flow channels with tubulin in imaging buffer: BRB80, 1 mM GTP, 0.1 mg/ml bovine serum albumin (BSA), 10 mM dithiothreitol, 250 nM glucose oxidase, 64 nM catalase, and 40 mM D-glucose. For low concentration tubulin experiments (50–500 nM), low retention tips and tubes were used.

## Turbidity bulk-phase microtubule polymerization assay

The formation of microtubule polymer was followed as the absorbance at 350 nm using a Cary 300 Bio UV-Visible spectrophotometer or Cary 3500 UV-Vis spectrophotometer (Varian Inc, Agilent Technologies, Santa Clara, CA). Tubulin was polymerized at 36 or 4°C in the presence or absence of CKAP2 in BRB80 containing 1 mM GTP and 1 mM $MgCl_2$.

## Microtubule nucleation assay

Flow channels were constructed as previously described but anti-TAMRA antibody was replaced with Anti-β3 tubulin antibody (BioLegend Poly18020 anti-Tubulin Beta 3). Microtubule nucleation was achieved by introducing 5–50 µM tubulin (1:17 molar ratio of Atto-633 labelled:unlabelled) in imaging buffer: into the flow channel. For CKAP2-positive experiments, freshly purified 0.2 µM CKAP2-mNG was introduced with 0.2–0.5 µM tubulin (1:17 molar ratio of Atto-633 labelled:unlabelled) in imaging buffer into flow channels. Tubulin was incubated 2 min, after which the total number of microtubules nucleated within one field of view (110 × 110 µm) was counted.

## Microtubule depolymerization assay

Assay channels were constructed as described above and incubated with 25% labelled TAMRA GMPCPP seeds. 0 or 1 µM CKAP2-mNG in imaging buffer without GTP was introduced into the channel. Channels were capped with nail polish to allow for longer imaging times. Microtubule seeds were imaged every 1 min during depolymerization for up to 10 hr.

## Profiles of CKAP2 tip tracking

Averaged fluorescence intensity profiles of CKAP2-mNG, Atto-633 tubulin (dynamic microtubule), and rhodamine tubulin (seed) were generated by taking two separate line scans, one extending from the dynamic microtubule lattice over the microtubule tip and one within the GMPCPP seed (number of line scans 6.5 nM [$n = 42$ seed, 54 tip], 12.5 nM [$n = 17$ seed, 26 tip]). One-dimensional line profile intensities for each separate channel were background subtracted, averaged, and normalized. The solid line represents the average intensity, and the shading represents intensity standard error.

## Preparation of paclitaxel-stabilized GDP microtubules

A polymerization mixture was prepared with BRB80 + 32 µM tubulin + 1 mM GTP + 4 mM $MgCl_2$ + 5% DMSO. The mixture was incubated on ice for 5 min, followed by incubation at 37°C for 30 min. The polymerized microtubules were diluted into prewarmed BRB80 + 10 µM paclitaxel, centrifuged at 110,000 rpm (199,000 × $g$) in an Airfuge (Beckman-Coulter), and resuspended in BRB80 + 10 µM paclitaxel (99.5+%, GoldBio).

## Curvature recognition

TAMRA-labelled paclitaxel microtubules were introduced into flow channels (*Bechstedt et al., 2014*). 5 nM CKAP2-mNG in imaging buffer with 10 µM paclitaxel without GTP was introduced into the channel. Quantification of the microtubule curvature, $\kappa$, and CKAP2-mNG signal on microtubules was analyzed using Kappa (*Mary and Brouhard, 2019*).

## Image analysis and software

For all in vitro data, image acquisition was controlled using MetaMorph (Molecular Devices) or ZEN 2.3 (Zeiss). Images were acquired from 2 s to 1 min intervals.

All images were processed and analyzed using Fiji (*Schindelin et al., 2012*) (ImageJ). If needed, prior to analysis images were corrected for stage drift using a drift correct script (Hadim). Microtubule dynamics were analyzed using kymographs (for cell data and in vitro data). Growth and shrinkage

rates were measured by manually drawing lines on kymographs and measuring the slope of growth or shrinkage.

Probability to nucleate was calculated by counting the number of seeds to nucleate a microtubule within 1 min over the total number of seeds within a field of view. Catastrophe frequency was measured by counting the total number of catastrophe events over the total time of all microtubule growth within a channel.

All functions were fitted and graphed with OriginPro2020 (OriginLab) or Python 3 (available at python.org) using a JupyterLab Notebook. Mean and standard deviation were calculated using Excel. Box plots represent the median, 25th and 75th percentile, whiskers are Q3 + 1.5 × IQR and Q1 − 1.5 × IQR; all data points including outliers are shown. Statistical analysis was performed using OriginPro2020. Images were linearly adjusted in brightness and contrast using Photoshop (Adobe). All figures were assembled using Illustrator (Adobe).

### Size exclusion chromatography

CKAP2-HUS alone, tubulin alone, and a CKAP2:tubulin mix were buffer exchanged and diluted to 4 µM into BRB80 + 60 mM KCl and incubated for 10 min at 4°C. 200 µl of each sample was then loaded onto a Superose 6 Increase 10/300 GL column (Cytiva) controlled with a BioRad NGC Quest 10 plus chromatography system.

### Sequence analysis

Amino acid sequences for CKAP2 (Uniprot accessions: hs Q8WWK9, mm Q3V1H1, xl A0A1L8HAI4), TPX2 (hs Q9ULW0, mm A2APB8, xl Q6NUF4), and chTOG (hs Q14008, mm A2AGT5, xl Q9PT63) were aligned using multiple and pairwise sequence alignments using Clustal Omega (*Sievers et al., 2011*) and visualized using MacVector 18.2.0.

## Acknowledgements

We thank Gary Brouhard as well as members of the Brouhard and Bechstedt labs for critical reading of the manuscript. We thank Kim Munro (CRSB, McGill University) for acquisition and analysis of CD spectroscopy experiments. Funding: The work was funded through CIHR PJT-156193 and NSERC RGPIN-2017-04649. TM was supported through a CRSB fellowship as well as FRQS doctoral fellowship.

## Additional information

### Funding

| Funder | Grant reference number | Author |
| --- | --- | --- |
| Canadian Institutes of Health Research | CIHR PJT-156193 | Susanne Bechstedt |
| Natural Sciences and Engineering Research Council of Canada | RGPIN-2017-04649 | Susanne Bechstedt |
| Fonds de Recherche du Québec - Santé | Doctoral Training | Thomas S McAlear |

The funders had no role in study design, data collection, and interpretation, or the decision to submit the work for publication.

### Author contributions

Thomas S McAlear, Data curation, Formal analysis, Investigation, Methodology, Software, Visualization; Susanne Bechstedt, Conceptualization, Funding acquisition, Investigation, Methodology, Project administration, Supervision, Writing - original draft

### Author ORCIDs

Thomas S McAlear ⬤ http://orcid.org/0000-0001-6097-0103
Susanne Bechstedt ⬤ http://orcid.org/0000-0002-4706-9975

Decision letter and Author response
Decision letter https://doi.org/10.7554/eLife.72202.sa1
Author response https://doi.org/10.7554/eLife.72202.sa2

## Additional files

### Supplementary files
• Supplementary file 1. Available plasmids used for Cytoskeleton-Associated Protein 2 (CKAP2) protein expression.

• Transparent reporting form

### Data availability
All data generated or analysed during this study are included in the manuscript and supporting files.

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
