## [Editor Report]

This study reports the first substantial in vitro characterization of how the purified microtubule-associated protein CKAP2 interacts with microtubules. CKAP2 strongly promotes their nucleation and polymerization and suppresses their depolymerization. This work may stimulate more investigations into the physiological significance of these remarkably strong in vitro activities.

---

## [Decision Letter]

**Decision letter after peer review:**

Thank you for submitting your article "The mitotic spindle protein CKAP2 potently increases formation and stability of microtubules" for consideration by *eLife*. Your article has been reviewed by 3 peer reviewers, one of whom is a member of our Board of Reviewing Editors, and the evaluation has been overseen by Anna Akhmanova as the Senior Editor. The reviewers have opted to remain anonymous.

Essential revisions:

1. To substantiate the discussion of the possible mechanism by which CKAP2 acts on microtubules, the authors should engage a bit more with the CKAP2 sequence/structure and the biochemistry of CKAP2's interaction with tubulin/microtubules. If the necessary instruments or reagents are available, the authors could perform experiments to measure the secondary structure of CKAP2 (CD spectroscopy), use CKAP2 fragments to clarify whether distinct parts of the molecule can be identified that bind selectively to microtubules or soluble tubulin, determine the stoichiometry of binding to soluble tubulin (size exclusion chromatography), or compare some of the observed behavior to that of other unstructured, highly charged proteins, such as for example TPX2 (if available).

2. The turbidity measurements in Figure 1 report a large change in turbidity upon addition of CKAP2, but it is unclear whether the turbidity increase is due to the polymerization of microtubules only or whether aggregates or microtubule bundles may contribute to increased turbidity. This possibility would at least need to be stated (if it cannot be excluded experimentally).

3. Some editorial changes are required. This concerns the comparison of the tubulin concentration used in in vitro experiments with physiological tubulin concentrations. Some statements about potential molecular mechanisms require checking to make sure that given the experimental evidence provided, conclusions are presented in a balanced manner. The Introduction and Discussion could be written in a more structured and coherent manner. The Discussion should come back to the reported phenotypes of CKAP2 depletions in cells and explain how the in vitro results reported here, help to understand these phenotypes/the function of CKAP2 in cells.

4. Please make sure to provide all necessary statistical information.

*Reviewer #1:*

The authors perform the first characterization of CKAP2 on microtubule nucleation and dynamic properties in vitro using purified proteins. They find that CKAP2 strongly promotes microtubule nucleation, strongly accelerates microtubule growth and suppresses catastrophes. It is interesting that this proteins seems to combine different properties of previously studied microtubule associated proteins. Its ability to promote microtubule nucleation and suppress catastrophes is to a certain extent similar to TPX2, its ability to accelerate microtubule growth is reminiscent of the microtubule polymerase chTOG/XMAP215. Particularly the strengths of the observed effects are striking. The quality of the measurements is high and the reported observations are clearly supported by the data. The mechanism underlying the various activities of CKAP2 remains however largely unknown at the current stage.

Some questions concerning the mechanism could be addressed experimentally, other could be discussed:

1. CKAP2, like TPX2, is predicted to be largely unstructured and both proteins are also highly positively charged (the latter property not being mentioned in the manuscript). The authors show that CKAP2 can recruit soluble tubulin to a GMPCPP microtubule and propose that this could be the basis for its growth accelerating effect, in a way similar to how chTOG/XMAP215 acts that consists however of well defined folded domains and is less charged. Doable experiments can provide answers to the following questions: does CKAP2 recruit free tubulin to microtubules more efficiently than the similarly charged TPX2? This could support the authors' idea of this tubulin recruitment being responsible for its growth speed accelerating effect, given that TPX2 does not increase growth speed. Is CKAP2 microtubule binding nucleotide-dependent as was observed for TPX2? This could give a hint about the specificity of binding to the microtubule lattice. Does CKAP2 bind tubulin with a defined stoichiometry similar to chTOG/XMAP215 (size exclusion chromatography)? This could support a specific tubulin binding mode.

2. Depletion of CKAP2 in cells has been reported to lead to multipolar spindles (Case et al., 2013). It would be useful to discuss how the findings here can help to explain this phenotype. TPX2 which is similar to CKAP2 in that it is predicted to be largely unstructured and highly positively charged is known to interact with multiple other proteins, having stimulated a debate regarding to which extent its effects on microtubule nucleation and dynamics are direct (as observed in experiments with pure proteins) or indirect (mediated by its interaction partners). Depletion of CKAP2 also seems to change the distribution of γ-tubulin and the association of centrosomes to poles in spindles. Could it be that the effects of CKAP2 on microtubule nucleation in cells are also indirect? Or that the effective CKAP2 concentration may be reduced due to interaction with other proteins which might explain why the effects in vitro appear more drastic than in cells?

The style of writing the Introduction and Discussion appears somewhat scattered. The organization of the points made could be more clearly organized and more coherent.

It would be useful to state clearly which CKAP2 isoform was produced, ideally by providing the accession number of the gene/protein.

Language: what is meant by "sharp" phosphorylation?

*Reviewer #2:*

The present study is a thorough in vitro analysis of purified (recombinant) CKAP2 on microtubule dynamics. The physiological relevance is overstated, in particular as cellular tubulin concentrations are way higher than assumed/used in this study (8 µM vs >20 µM).

Main Concerns:

1) Physiological relevance and tubulin concentrations

The manuscript emphasises the physiological relevance of CKAP2, often refers to physiological concentrations of tubulin and CKAP2 and claims that experiments are done at physiological relevant concentrations. Unfortunately, 8 µM tubulin is significantly below cellular tubulin concentrations. The tubulin concentration in living cells is usually higher (up to 24 µM, Hiller and Weber, 1987), same in *Xenopus* egg extracts (20-25 µM, Gard and Kirschner, 1987, Good et al. 2013, …) and in *C. elegans* it is even thought to be around 50 µM (Baumgart et al. 2019, Saha et al. 2016).

In line 276, the authors reference Wühr et al. 2014 as source for "the tubulin concentration was determined to be ~ 7.5 μM". Unfortunately, this is incorrect. This is the concentration of a single tubulin isoform, the authors would need to some up all identified tubulin isoforms.

Either the authors need to correct for a relevant tubulin concentration (in the text as well as the experiments) or show the thorough in vitro characterisation as it is without claiming physiological relevance.

2) Statistics

It would be preferable to see triplicates for each experiment. Most supplementary figures completely lack information on statistics

3) Microtubule- and tubulin-binding domain

Line 205: "CKAP2 therefore possesses a distinct microtubule lattice as well as likely one or multiple tubulin binding sites. Recruitment of tubulin by CKAP2 could therefore play a role in promoting microtubule nucleation and growth." I do not think that this rather strong statement is backed up by the experiments shown. Tubulin binding of CKAP2 could for example be shown directly by size exclusion chromatography.

Line 62: "lack of any molecular understanding of this protein"

Please, be more specific. There is some molecular understanding, for example KEN-box mediated CKAP2 degradation. "No understanding of effect on microtubules" would be fair.

Line 94: "enhances tubulin polymer formation"

Would be helpful to see entire field of views per condition.

Line106: "previously observed"

Reference?

Line 147: "we turned to an assay"

It is the very same assay, just different data analysis.

Line 192/193 and Figure 1(a)

What is the relevant tubulin alone conc. to compare this to?

Line 285: "Notably, we purify CKAP2 from *E. coli* in an unphosphorylated state."

This is very likely to be true (based on *E. coli* expression and Coomassie). However, this is a strong statement that either needs verification or should be toned down.

Figure 1 (a)

Dosztanyi et al., 2005 ref for IUPred, no Ref for PONDOR

Suppl. Figure 1 (a) and (b)

Would be nice to see these at the same scale! Are these triplicates?

Figure 3 (a)

- Why is this not a graded response but rather a switch-like from 12.5 nM to 25 nM?

- Title: "and catastrophe" not shown in this figure.

Figure 3 (d)

- Are the intensity profiles scaled equally?

Figure 4(a)

Colors?

Suppl. Figure 4(c)

What is "cumulative microtubule survival"?

General comment

1) Would be helpful to see the zoom-ins (for example Figure 2C and S2B) side-be-side in the main figures.

2) On the discussion and comparison with XMAP215

– Line 314/315: "Unlike a polymerase, CKAP2 does not catalyze the reverse reaction of tubulin polymerization in the absence of tubulin substrates." However, in Figure 2e, the authors show that at low (physiological) concentrations the apparent off-rate is 6-9/s.

– For XMAP215 microtubule growth rates level off at around 200 nM. At this concentration XMAP215 is expected to saturate the microtubule (+)-end. What about CKAP2? Figure 2b implies no saturation in growth velocity. Would be an interesting point for discussion.

*Reviewer #3:*

This goal of this manuscript was to obtain in vitro data to characterize how CKAP2 affects microtubule stability and dynamics.

Strengths:

There are three main strengths to the manuscript.

– First, novelty – the work describes the microtubule regulatory activity of a relatively uncharacterized protein. Microtubules have been heavily studied for some time, so this is not a common event.

– Second, the results themselves are striking, especially the very large effect on promoting polymerization

– Third, the helpful initial steps toward ascribing mechanisms, such as the data indicating that CKAP2 can bind a microtubule and an unpolymerized ab-tubulin simultaneously.

Weaknesses:

– Figure 1 is constructed to suggest that there is ~50-fold more microtubules in the +CKAP2 assay compared to control. But some kind of control is lacking to substantiate this – either a taxol or GMPCPP reaction to indicate the signal obtainable from maximal assembly, or some kind of imaging to demonstrate that the product of the CKAP2 reaction is microtubules (and not, say, bundles of microtubules that would scatter more and therefore give a stronger signal).

– Lack of engagement with CKAP2 sequence and/or structure. Some of the results raise obvious questions about potential CKAP2 sequence elements, but does not address them. Figure 1 shows two server predictions that CKAP2 is intrinsically disordered, but there are not any solution measurements (possibilities: CD spectroscopy to assess secondary structure content, size exclusion chromatography to assess apparent size, limited proteolysis to identify or show lack of resistant fragments of the protein) to substantiate. Figure 5 shows that CKAP2 can bind microtubules and unpolymerized tubulin at the same time, suggesting separable binding sites. Is there an identifiable basic region that might mediate microtubule binding? Does a multiple sequence alignment provide any hints about more conserved regions that might underlie these activities? 50-fold increase in apparent on-rate constant is quite striking, but there is little discussion/speculation about possible mechanisms.

Appraisal:

This is high-quality, interesting work with only minor weaknesses. Goals to characterize the activity of CKAP2 on microtubules were clearly met. The most striking effects are the promotion of nucleation and the very large increase in the apparent on-rate constant for elongation. Demonstration of separable binding sites for microtubules and ab-tubulin and lack of effect on depolymerization are useful but not as unambiguously established.

Impact:

The work will likely generate strong interest – it is not common to read about previously uncharacterized microtubule regulators, and the results obtained provide useful benchmarks for future studies aimed at a better understanding of mechanisms by which this protein acts.

– This is a question of taste, but the introduction raised a number of questions (regulation of microtubule growth and nucleation in spindles, whether all relevant regulatory factors had been identified) that the manuscript does not answer. I recommend condensing or at least reframing some of this more open-ended material.

– In Figure 1D, I think it would be helpful to have some sort of 'maximal assembly' control (GMPCPP, taxol, or other). This, or an experiment to investigate the state of the assembly products (individual microtubules vs bundles vs other aggregated forms) would give context for thinking about the comparison of turbidity signals. Experiments are not strictly necessary here given the results that follow, so this comment could also be addressed with a few text modifications.

– For Figure 2C and more generally – the difference in activity is so strong that the +CKAP2 measurements look like almost vertical lines. The authors show enlarged plots in supplemental materials, but I think it would be nicer to have these as insets in the main figure.

– It is somewhat hard to tell from Sup. Figure 2A, but it appears that CKAP2 might show plus-end specific activity (minus end growth rate not affected). If true this seems worthy of a comment somewhere in the text.

– Sup. Figure 3A shows an interesting result – that even at ~5-fold+ superstoichiometry over tubulin, CKAP2 increases the probability of 'firing' from a seed. This has implications for the affinity or mode of binding to ab-tubulin and might be worth commenting on.

– Around line 205 the authors describe data showing CKAP2 can bind simultaneously to a microtubule and to ab-tubulin. This suggests separable binding sites. If possible, it would be nice to know if there are hints in the sequence about this – identifiable basic region(s), and/or especially conserved regions that may hint at binding sites. Identifying such regions could inform future work and support speculation about possible mechanisms in the present manuscript.

– The +CKAP2 kymograph in Figure 4 looks to show an elongation rate that slows over time. Is that typical? In the legend to Figure 4, I assume n refers to microtubules (not catastrophes). Could the authors also state how many catastrophes they observed, e.g. 7 catastrophes observed for 234 microtubule growth episodes examined, or something like that?

­– For Figure 5, is it possible to look for effects on shrinking rate at lower concentrations of CKAP2, where the lattice is not coated? This is not essential but would be nice to know.

– Adding some speculation about possible mechanisms, or at least a more substantive comparison between magnitude of promoting growth rates, would help put the work into a better context. The paragraph at line 323 makes a beginning on this.

---

## [Author Response]

Essential revisions:1. To substantiate the discussion of the possible mechanism by which CKAP2 acts on microtubules, the authors should engage a bit more with the CKAP2 sequence/structure and the biochemistry of CKAP2's interaction with tubulin/microtubules. If the necessary instruments or reagents are available, the authors could perform experiments to measure the secondary structure of CKAP2 (CD spectroscopy), use CKAP2 fragments to clarify whether distinct parts of the molecule can be identified that bind selectively to microtubules or soluble tubulin, determine the stoichiometry of binding to soluble tubulin (size exclusion chromatography), or compare some of the observed behavior to that of other unstructured, highly charged proteins, such as for example TPX2 (if available).

We are engaging in the text with CKAP2’s primary sequence, charge, and domain composition and are showing sequence alignments between human, mouse, and frog CKAP2; annotate known domains; and show comparisons between CKAP2, TPX2, and chTOG for evolutionary conservation (new Figure S1).

We have confirmed the prediction of the largely disordered nature of CKAP2 in solution by circular dichroism (data in new Figure 1B).

We are including size exclusion chromatography data showing no interaction of CKAP2 with tubulin in solution (new Figure 5A).

We have started structure-function studies of CKAP2, but some fragments are not purifying or are aggregated upon purification, such that we cannot provide a comprehensive answer to which part of the molecule interacts with tubulin/microtubules at this point.

A behavior we have observed for CKAP2 with regards to tubulin binding, but are not including in the manuscript, is that CKAP2 undergoes liquid-liquid phase separation in low salt or PEG-containing buffers as expected for an IDP. Tubulin partitions into these condensates and in the presence of GTP ‘aster’-like microtubule arrays form (see Author response image 1). As we (and other labs) have observed this behavior with many other MAPs, we are not sure this behavior is relevant for the in vivo function of CKAP2 and had therefore not included this data in the manuscript. Though, in light of the reviewer request to compare CKAP2 with TPX2 and the recent TPX2 publications from Sabine Petry’s lab reporting this phenomenon for TPX2 (King et al. Nat. Comm. 2020, Setru et al., Nat. Physics 2021, and Safari et al. JBC 2021), we could offer to include these observations if recommended by reviewers and editors.

**Author response image 1. sa2fig1:** 

2. The turbidity measurements in Figure 1 report a large change in turbidity upon addition of CKAP2, but it is unclear whether the turbidity increase is due to the polymerization of microtubules only or whether aggregates or microtubule bundles may contribute to increased turbidity. This possibility would at least need to be stated (if it cannot be excluded experimentally).

We have toned down the description of the turbidity data and acknowledge the contribution of bundling to the large increase in turbidity caused by CKAP2.

3. Some editorial changes are required. This concerns the comparison of the tubulin concentration used in in vitro experiments with physiological tubulin concentrations. Some statements about potential molecular mechanisms require checking to make sure that given the experimental evidence provided, conclusions are presented in a balanced manner. The Introduction and Discussion could be written in a more structured and coherent manner. The Discussion should come back to the reported phenotypes of CKAP2 depletions in cells and explain how the in vitro results reported here, help to understand these phenotypes/the function of CKAP2 in cells.

The concerns for tubulin concentration seem to stem from reviewer 2’s comments:

“The manuscript emphasises the physiological relevance of CKAP2, often refers to physiological concentrations of tubulin and CKAP2 and claims that experiments are done at physiological relevant concentrations. Unfortunately, 8 µM tubulin is significantly below cellular tubulin concentrations. The tubulin concentration in living cells is usually higher (up to 24 µM, Hiller and Weber, 1987), same in Xenopus egg extracts (20-25 µM, Gard and Kirschner, 1987, Good et al. 2013, …) and in *C. elegans* it is even thought to be around 50 µM (Baumgart et al. 2019, Saha et al. 2016).

In line 276, the authors reference Wühr et al. 2014 as source for "the tubulin concentration was determined to be ~ 7.5 μM". Unfortunately, this is incorrect. This is the concentration of a single tubulin isoform, the authors would need to some up all identified tubulin isoforms.”

We have taken both the CKAP2 and tubulin concentration approximations from Wuehr et al. 2014. Unlike stated in one review, we had added the different tubulin isoforms (see Author response table 1 and Table S2/S3, Wuehr et al. 2014). While we do acknowledge that tubulin concentrations with other methods and in other systems have been determined to be anywhere between 8 µM and 50 µM, it simply does not seem prudent to take the CKAP2 concentration estimation from one study and pick a tubulin concentration from a different study using a different method in a different organism. Moreover, we show in Figure 2 C/D a linear relationship of growth rates with tubulin concentrations across three different CKAP2 concentrations. We have therefore explored concentration spaces for both tubulin and CKAP2.

**Author response table 1. sa2table1:** 

Published concentration for *Xenopus* (Table S2/S3, Wuehr et al. 2014)					
Α Tubulins	ng	Β Tubulins	ng		
TUBA4A	Tubulin α-4A chain	4741.14	TUBB	Tubulin β chain	5931.21
TUBA3C	Tubulin α-3C/D chain	1686.89	TUBB2B	Tubulin β-2B chain	1357.19
TUBA1B	Tubulin α-1B chain	1143.46	TUBB4B	Tubulin β-4B chain	1173.38
TUBA1A	Tubulin α-1A chain	790.93			
TUBA1C	Tubulin α-1C chain	439.50			
		8801.91			8461.78

In addition, we would like to point out that measurements of microtubule polymerases and nucleation factors have been performed at 7.5 µM tubulin (chTOG+TPX2, Roostalu et al. 2016), 15 µM (XMAP215, Kinoshita et al. 2001) and 4.5 µM (XMAP215, Brouhard et al. 2008), such 8 µM used in our experiments is perfectly in line with the leading literature in the field.

To accommodate any strong opinions about the absolute tubulin concentration in cells, we have taken all previous statement for cellular tubulin concentrations out of the manuscript and refer to measurements around 50-70 nM CKAP2 and 8 µM tubulin as “quasi-physiological” as well as “8 µM tubulin, the lower end of estimated cellular tubulin concentrations”. We would like to keep at some reference like quasi-physiological to refer to our measurements at 8 µM in the manuscript to contrast these to the measurements at 50 nM tubulin, which we consider at least 2 orders of magnitude below physiological concentration ranges.

We have made changes to both the Introduction and the Discussion sections to provide a better structure and provide a better link between our in vitro findings and the cellular phenotypes described in the literature.

4. Please make sure to provide all necessary statistical information.

We are now providing statistical information for all data and include Tukey’s test and one way ANOVA to compare mean values. New image analysis quantification of CKAP2 end tracking presented in two new Figure panels in Figure 5 C.